# Update on Antioxidant Therapy with Edaravone: Expanding Applications in Neurodegenerative Diseases

**DOI:** 10.3390/ijms25052945

**Published:** 2024-03-03

**Authors:** Toru Yamashita, Koji Abe

**Affiliations:** 1Department of Neurology, Okayama University Graduate School of Medicine, Dentistry and Pharmaceutical Sciences, 2-5-1 Shikata-cho, Okayama 700-8558, Japan; 2Department of Neurology, National Center of Neurology and Psychiatry, Tokyo 187-8551, Japan

**Keywords:** edaravone, free radical scavenger, stroke, amyotrophic lateral sclerosis, Alzheimer’s disease, placental ischemia

## Abstract

The brain is susceptible to oxidative stress, which is associated with various neurological diseases. Edaravone (MCI-186, 3-methyl-1 pheny-2-pyrazolin-5-one), a free radical scavenger, has promising effects by quenching hydroxyl radicals (∙OH) and inhibiting both ∙OH-dependent and ∙OH-independent lipid peroxidation. Edaravone was initially developed in Japan as a neuroprotective agent for acute cerebral infarction and was later applied clinically to treat amyotrophic lateral sclerosis (ALS), a neurodegenerative disease. There is accumulating evidence for the therapeutic effects of edaravone in a wide range of diseases related to oxidative stress, including ischemic stroke, ALS, Alzheimer’s disease, and placental ischemia. These neuroprotective effects have expanded the potential applications of edaravone. Data from experimental animal models support its safety for long-term use, implying broader applications in various neurodegenerative diseases. In this review, we explain the unique characteristics of edaravone, summarize recent findings for specific diseases, and discuss its prospects for future therapeutic applications.

## 1. Introduction

The brain, which is rich in lipids and exhibits high oxygen consumption, is susceptible to damage via oxidative stress. Briefly, oxidative stress is caused by an imbalance between the production and accumulation of reactive oxygen species (ROS) in cells and tissues and the ability of a biological system to detoxify these reactive products [1]. ROS contribute to several physiological processes (e.g., cell signaling) [2] and are generated as byproducts of oxygen metabolism under normal conditions. Nevertheless, environmental stressors and xenobiotics can contribute to a significant increase in ROS production, resulting in cellular and tissue damage. Oxidative stress has been implicated in neurodegenerative diseases, including amyotrophic lateral sclerosis (ALS), Parkinson’s disease, Alzheimer’s disease (AD), Huntington’s disease, depression, and multiple sclerosis. Furthermore, it plays an important role in the pathogenesis of acute ischemic stroke [3,4]. Free radical formation and subsequent oxidative damage may be a factor in stroke severity [5]. Therefore, several antioxidants with demonstrated or predicted beneficial effects against oxidative stress and stroke have recently been reported [6,7,8,9].

Edaravone (MCI-186, 3-methyl-1 pheny-2-pyrazolin-5-one) was first established in Japan as a free radical scavenger (Figure 1). It was initially approved for the treatment of acute ischemic stroke in Japan, where it is manufactured under the brand name Radicut. It has been used in clinical practice for the treatment of acute cerebral ischemia and ALS owing to its antioxidative and anti-inflammatory effects. Edaravone has also been approved for use in Japan, South Korea, and the United States for the treatment of ALS. It has been approved in the United States under the brand name Radicava by the U.S. Food and Drug Administration (FDA).

In this study, using PubMed, we searched the literature for studies related to edaravone for a comprehensive review of its development and applications to various diseases. We introduce the unique characteristics of edaravone, summarize recent findings for various diseases (Table 1 and Table 2), and discuss prospects for future therapeutic applications.

## 2. Edaravone Development

In the 1970s, it was proposed that oxygen-free radical species damage cellular lipid membranes and play an important role in the pathophysiology of various diseases, including ischemic stroke [3,4]. In infarcted brains, ischemic energy depletion increases cytosolic Ca^2+^ levels and activates phospholipase A2 through pump failure and cell depolarization. Phospholipase A2 releases free fatty acids, particularly arachidonic acid, from the cell membrane. Arachidonic acid is a precursor for the synthesis of various bioactive lipid mediators, including prostaglandins, thromboxanes, and leukotrienes. These lipid mediators play important roles in inflammation, leading to the bursting of free radicals in the ischemic penumbra [27].

To establish new and effective drugs for stroke, many researchers have sought free radical scavengers without side effects, such as narcotic effects or inhibitory effects on brain metabolism. Phenolic compounds promote prostaglandin metabolic activity by scavenging the most toxic reactive oxygen species, hydroxyl radicals [28]. Edaravone, screened based on effects on prostaglandin synthesis, is a derivative of 2-pyrazolin-5-one and shows similar activity to that of phenol. Edaravone has promising effects by quenching hydroxyl radicals (∙OH) and inhibiting ∙OH-dependent and ∙OH-independent lipid peroxidation (Figure 1) [10]. It also exhibits biphasic water-soluble and liposoluble properties, with a cLogP value of 1.33. cLogP represents the logarithm of the partition coefficient between n-octanol and water (log(c_octanol_/c_water_)) and is an established indicator of hydrophilicity.

Edaravone showed inhibitory effects on water-soluble and fat-soluble radical-induced peroxidative systems, which differed from the inhibitory effects of vitamin E (cLogP = 9.96) and vitamin C (cLogP = −2.15) [29]. Under physiological conditions, about half of edaravone exists as edaravone anions, which donate electrons to various types of free radicals and scavenge free radicals [7].

## 3. Neuroprotective Effect of Edaravone on Acute Stroke

As noted earlier, activation of the arachidonic acid cascade is a major cause of edema and tissue injury in cerebral ischemia. Watanabe et al. injected arachidonic acid directly into the cerebral cortex of rats and evaluated its anti-edema effect. Edaravone administration (0.1–3.0 mg/kg i.v.) suppressed brain swelling markedly 24 h after arachidonic acid injection [11]. Abe et al. administered edaravone to a rat model of transient middle cerebral artery occlusion (tMCAO) and evaluated the effect on post-stroke brain edema. In this model, the water content, reflecting the breakdown of the blood–brain barrier, increased significantly after 3 and 6 h of ischemia. Further increases were observed after 3 h of ischemia and 3 h of reperfusion. As a result, edaravone markedly inhibited ischemia and post-ischemic brain swelling [12]. In addition, post-ischemic administration of edaravone reduced the size of the infarct considerably and improved neurological deficits 1 day after tMCAO [10]. Kawai et al. reported that edaravone markedly inhibited the accumulation of the nucleic acid oxidation product 8-oxo-7,8-dihydro-2-deoxyguanosine (8-oxodG) and the continuous inflammatory response in peri-infarct lesions in a mouse stroke model [13]. These results indicate that edaravone reduces ischemic brain damage in animal models of stroke by inhibiting free radical generation and suppressing the arachidonic acid cascade and inflammatory reactions (Table 1). In addition, studies have shown that edaravone administration can increase nitric oxide-mediated vasodilation through a decrease in oxidative stress, suggesting that edaravone exerts a brain-protective effect by maintaining cerebral blood flow [30,31].

### 3.1. Clinical Trials of Edaravone in Patients with Acute Stroke 

In a multicenter clinical trial, edaravone reduced disability in humans 90 days after acute ischemic stroke without serious side effects [20]. As a result, edaravone is widely used in Japan as an effective neuroprotective agent for patients with acute stroke. Edaravone is available in an intravenous (IV) formulation. Of note, acute renal failure occurs in elderly patients with renal dysfunction [32]. Accordingly, renal function should be monitored continuously in patients receiving edaravone.

In a multicenter, double-blind, randomized clinical trial involving 1165 patients with acute stroke, Xu et al. investigated the neuroprotective effects of a combination of edaravone and dexborneol (a naturally occurring terpene and bicyclic organic compound). The edaravone-dexborneol group exhibited significantly better functional outcomes than those of the edaravone group, especially in female patients, supporting the potential benefits of edaravone-dexborneol in enhancing functional recovery after stroke [21] (Table 2).

### 3.2. Vascular Protective Effect of Edaravone after Thrombolytic Therapy with Tissue Plasminogen Activator

Edaravone was developed for the treatment of acute stroke. The advent of thrombolytic therapy with tissue plasminogen activator (tPA) for stroke highlights the importance of free radical scavenger therapy in augmenting the role of free radical scavengers in clinical practice. Restoration of cerebral blood flow with tPA can ameliorate ischemic brain injury [33], and intravenous tPA is used worldwide as a thrombolytic therapy. However, delayed reperfusion with tPA can cause hemorrhagic transformation (HT) [34], mainly through the activation of matrix metalloproteinase-9 (MMP-9). Therefore, the clinical application of tPA remains limited.

Yamashita et al. used a spontaneous hypertension model of tMCAO to test the efficacy of edaravone in preventing HT. tPA alone significantly worsened survival compared with that in the vehicle control group. However, the combination of edaravone and tPA significantly increased survival, improved motor function, and dramatically reduced HT [14]. In addition, edaravone treatment suppressed MMP-9 expression in and around cerebral microvessels, inhibited basement membrane protein degradation, and prevented microvascular dissection. These results indicate that edaravone protects the integrity of cerebral microvessels by protecting the basement membrane from excess free radicals and MMP-9, leading to a subsequent reduction in HT and improved survival and neurological outcomes.

Another free radical scavenger, NXY-059, potentially inhibited symptomatic HT after tPA treatment [35]; however, this effect was not replicated in a subsequent study [36]. NXY-059 is highly soluble with a cLogP of −2.09. Edaravone, on the other hand, is biphasic, being both water- and fat-soluble (cLogP = 1.33). It can easily cross the blood–brain barrier and enter the brain parenchyma and cerebral fluid [29]. This unique chemistry of edaravone may favor delivery to the basement membrane since the sites most susceptible to damage by free radicals are outside the vascular endothelium (e.g., the basement membrane). Thus, combination therapy with edaravone and tPA is a promising therapeutic strategy for patients with acute stroke, not only reducing infarct size but also minimizing fatal HT.

When edaravone was combined with tPA thrombolysis, early recanalization was more likely to be achieved [22]. In the Yamato study conducted in Japan, there was no difference in the preventive effect of edaravone against hemorrhagic stroke or recanalization between patients who received edaravone before or during thrombolytic therapy and those who received it afterward [23] (Table 2). In contrast, one retrospective cohort study showed that edaravone in combination with endovascular therapy significantly improved ADL at discharge, reduced mortality, and prevented hemorrhagic stroke [37]. Therefore, further clinical studies are warranted.

Free radicals play an important role in cerebral hyperperfusion syndrome after carotid endarterectomy, as observed during reperfusion with tPA therapy. Ogasawara et al. [38] reported that pretreatment with edaravone prevents the occurrence of cerebral hyperperfusion syndrome after carotid endarterectomy in patients with ipsilateral internal carotid artery stenosis. 

## 4. Therapeutic Effect of Edaravone on ALS

ALS is a neurologically intractable disease that usually develops in middle age or later and causes selective degeneration of motor neurons, resulting in weakness of limb muscles, muscle atrophy, dysarthria, dysphagia, and eventually respiratory failure due to respiratory muscle paralysis and death within 3–5 years. In 1993, a point mutation in the Cu/Zn SOD (*SOD1*) gene was discovered to be the causative gene of familial ALS (fALS) [39,40], accounting for approximately 30% of all fALS cases. Interestingly, SOD1 detoxifies superoxide (O_2_^−^), suggesting that free radicals play an important role in the pathogenesis of fALS [41]. It has also been reported that misfolded SOD1, with an abnormally folded structure, is involved in the pathogenesis of sporadic ALS (sALS), which accounts for the majority of ALS cases, and free radical damage caused by abnormal SOD1 is presumed to be closely involved in the pathogenesis of sALS [42]. In the brains of patients with ALS, there is evidence of decreased expression of glutathione peroxidase, an enzyme that detoxifies ROS and SOD1, increased expression of 8-OHdG, an indicator of DNA peroxidation, increased expression of lipid peroxidation markers, and nitrotyrosine expression [43]. 

Ito et al. performed a randomized, double-blind trial of edaravone treatment in G93A mutant SOD1 transgenic mice (ALS mice). They found that edaravone treatment significantly slowed the motor decline in the mouse model of ALS [15]. In ALS model mice, it was also found that 8-OHdG-positive cells appear in the perinuclear area of the spinal cord, even before the loss of motor neurons in the anterior nucleus of the spinal cord and the appearance of clinical symptoms. Furthermore, 8-OHdG expression increased in parallel with motor neuron degeneration and a loss of motor neurons. In addition, recent studies using ^62^Cu-ATSM PET imaging have shown a significant increase in accumulation in the bilateral motor cortex and right parietal cortex in patients with ALS compared with those in controls. The degree of accumulation is correlated with disease severity in patients with ALS [44]. These findings suggest that both wild-type and mutant forms of the SOD1 protein can interact with free radicals, and this is associated with the pathological mechanisms of ALS. Edaravone probably exerts therapeutic effects not by acting directly on SOD1 but by inhibiting downstream free radicals and associated inflammatory responses. 

### 4.1. Various Inflammation-Related Reactions Induced via Free Radical Damage

The Kelch-like ECH-associated protein 1 (Keap1)/Nuclear erythroid 2-related factor 2 (Nrf2) system has attracted attention as a protective mechanism against oxidative stress [45,46]. We found that Keap1 is down-regulated and Nrf2 is up-regulated in the anterior horn of the spinal cord in early-onset ALS, and this trend becomes more pronounced as the disease progresses. In vivo Nrf2 imaging analyses have revealed accelerated oxidative stress in spinal motor neurons and lower limb muscles of Nrf2/G93A mice according to disease progression. These symptoms were significantly alleviated by edaravone treatment, accompanied by clinical improvement (Figure 2). In addition, activation of MMPs, which are associated with neurovascular unit (NVU) damage, occurs before the onset of ALS in the anterior horn of the spinal cord of ALS model mice. This suggests that free radical damage, MMP activation, and NVU disruption play a major role in the pathogenesis of ALS [16] (Table 1).

### 4.2. Application of Edaravone to ALS

As mentioned above, free radicals are presumed to be involved in the pathogenesis of ALS, and edaravone was developed in Japan to treat the acute phase of cerebral infarction. To explore the therapeutic potential of edaravone, Yoshino et al. conducted a Phase II clinical trial of edaravone in patients with ALS [24]. Edaravone at 60 mg/day for 24 weeks inhibited the deterioration of the ALS Functional Rating Scale (ALSFRS-R) and respiratory function and reduced the expression of 3-nitrotyrosine, a marker of oxidative stress, in the spinal fluid. Based on these results, the first Phase III clinical trial was conducted in Japan in April 2006 in patients with ALS, revealing that 60 mg/day edaravone treatment was more effective than the placebo in preventing a decline in hand muscle strength (pinch strength). However, the primary endpoint, ALSFRS-R, did not reach a significant difference from that in the placebo group, and a second Phase III trial was conducted in 2012 involving patients with ALS severity class 1 or 2 and with an effort lung capacity of 80% or greater. Edaravone administration inhibits ALS progression [25,26] (Table 2).

Consequently, in June 2015, edaravone was approved as the world’s first therapeutic drug to show efficacy in inhibiting ALS progression. In addition, a retrospective study showed that long-term edaravone administration improves the survival of patients with ALS [47]. Edaravone therapy for ALS has been approved in South Korea and the United States, and its therapeutic effects have attracted considerable attention worldwide. In ALS treatment, edaravone was also administered via intravenous infusion. However, because long-term intravenous infusion is burdensome for patients with ALS, an oral suspension of edaravone was developed and is now used in clinical practice.

## 5. Therapeutic Effect of Edaravone on Alzheimer’s Disease

AD and cerebrovascular disease often coexist in patients with dementia, and approximately 90% of patients with AD over 75 years of age also have cerebrovascular disease. Chronic cerebral hypoperfusion (CCH) is also prevalent in elderly patients with AD and contributes to the pathophysiology of AD, involving amyloid-β (Aβ) overproduction, clearance impairment, Tau-hyperphosphorylation, neuroinflammation, oxidative stress, and neuronal loss. Despite recent progress, efficient disease-modifying therapeutics for AD remain limited. Considering the limitations of targeting a single protein or pathway in complex diseases, novel therapeutic strategies that address multiple key pathways are needed. Oxidative stress is a common manifestation of AD and CCH and accelerates their progression. 

Ueno et al. investigated the protective effects of edaravone in a rat model of chronic hypoperfusion that underwent bilateral common carotid artery ligation. They discovered that edaravone treatment enhanced endothelial nitric oxide synthase and significantly suppressed oxidative damage and oligodendrocyte loss in the cerebral white matter [17]. We also assessed the therapeutic potential of edaravone in a mouse model of AD (APP23) with CCH. In comparison with untreated mice with AD and CCH at 12 months, edaravone treatment significantly improves motor and cognitive deficits, reduces Aβ/pTau accumulation, and mitigates neuronal loss, oxidative stress, and neuroinflammation (Figure 3). We also observed that edaravone treatment significantly reduced CCH-induced white matter lesions in the corpus callosum of APP23 mice. These improvements include enhanced white matter integrity, increased proliferation of oligodendrocyte progenitor cells, mitigated endothelium/astrocyte unit dysfunction, and reduced neuroinflammation and oxidative stress [18] (Table 1). These findings suggest that edaravone has clinical and pathological benefits in this mouse model, highlighting its potential as a therapeutic agent for AD with CCH by targeting multiple key pathways involved in the pathogenesis of the disease.

## 6. Therapeutic Effect of Edaravone on Placental Ischemia

Preeclampsia (PE) is a pregnancy-related hypertensive disorder characterized by hypertension with or without proteinuria. It is associated with an increased risk of fetal and maternal morbidity and mortality, presenting a major health problem. Accumulated evidence suggests that reduced uteroplacental perfusion and placental ischemia play critical roles in this disorder [48]. Atallah et al. explored the effect of edaravone in a mouse model of reduced uterine perfusion pressure (RUPP)-induced placental ischemia. RUPP was induced on gestational day 13, and edaravone (3 mg/kg) was administered from day 14 to the day of sacrifice on day 18 of gestation. The findings revealed that edaravone significantly lowered maternal blood pressure, increased fetal survival rate, and enhanced fetal length, weight, and fetoplacental ratio compared with those in the RUPP group. In addition, edaravone mitigated fetal morphological abnormalities and improved ossification of the fetal endoskeleton. Furthermore, edaravone positively influenced the histopathological structure of the maternal kidney and heart and reduced elevated blood urea and creatinine levels (Figure 4) [19] (Table 1).

They also showed that RUPP surgery in pregnant mice induced structural changes and neurodegeneration and increased the expression of pro-inflammatory markers in fetal brains. Microglial and astrocyte activation and elevated Hif-1α and iNOS indicated oxidative stress. In contrast, edaravone administration demonstrated neuroprotective effects by reducing inflammation, restoring neuronal structures, and decreasing oxidative stress. These results suggest that edaravone is a potential adjuvant therapy to safeguard the developing fetal brain [19]. 

## 7. Conclusions and Future Prospects

Edaravone was initially developed in Japan as a neuroprotective agent for acute cerebral infarction and was later applied clinically to treat ALS, a neurodegenerative disease. The results of in vitro and in vivo animal studies indicated that edaravone has therapeutic effects on a wide range of diseases related to oxidative stress, including AD and placental ischemia, and its therapeutic range is expected to expand further. A suspension solution of edaravone has already been developed for patients with ALS, enabling long-term use in the chronic phase. Its application to various neurodegenerative diseases, including AD, may be in sight.

## Figures and Tables

**Figure 1 ijms-25-02945-f001:**
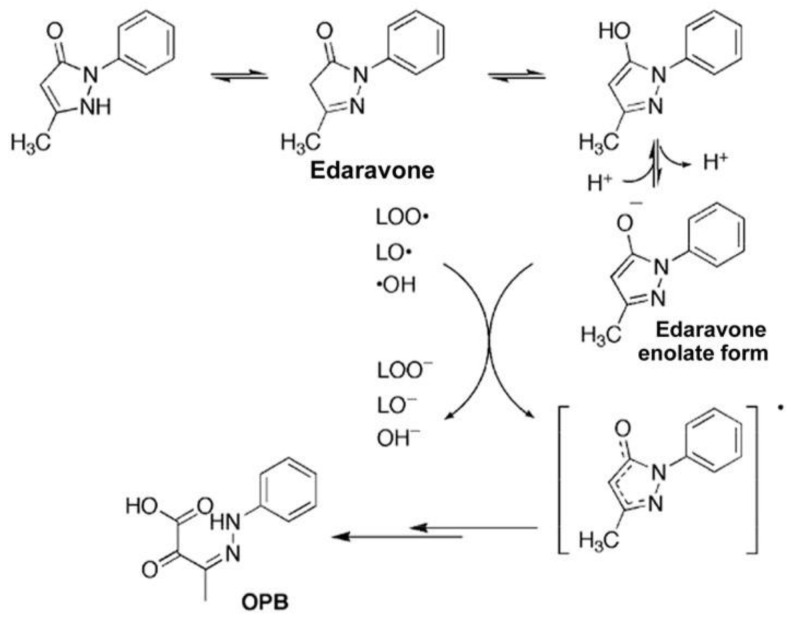
Reaction mechanism of edaravone with free radicals (revised from Nakagawa et al. 2006 [10]). The enolate form of edaravone interacts with both peroxyl radicals (LOO·) and hydroxy radicals (·OH) to form stable oxidation products (2-oxo-3-(phenylhydrazono)-butanoic acid; OPB).

**Figure 2 ijms-25-02945-f002:**
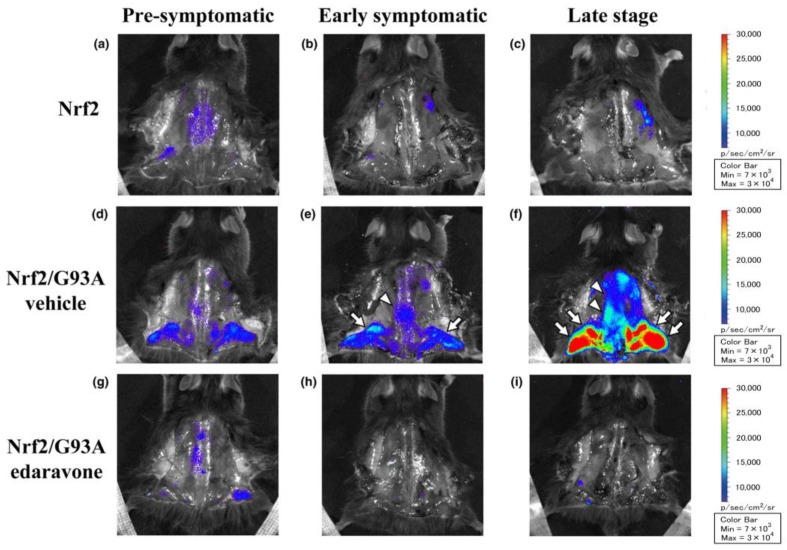
In vivo oxidative stress imaging of Nrf2/ALS mice (revised from Ohta et al., 2019 [16]). (**a**–**f**) Note strong Nrf2 signals in the spine (**e**, arrowhead) and lower limbs (**e**, arrows) of vehicle at the early symptomatic stage (14 weeks of age) with a further emphasis at the late stage (18 weeks of age, **f**, arrowheads and arrows). Edaravone treatment markedly suppressed Nrf2 expression (**g**–**i**).

**Figure 3 ijms-25-02945-f003:**
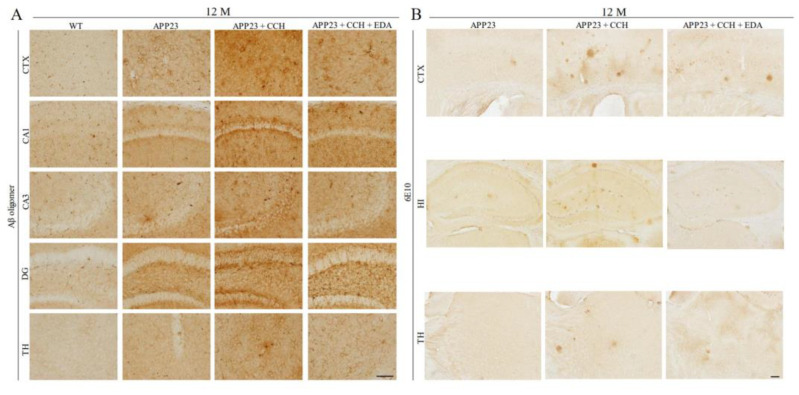
Representative photomicrographs of Aβ oligomer (**A**) and all forms of Aβ burden (**B**) in the CTX, CA1, CA3, DG, and TH at 12 M (revised from Feng et al. [18]). Edaravone treatment reduces the expression of Aβ oligomer and Aβ burden in APP23 + CCH mice. Scale bar = 50 µm. Abbreviation: Aβ, amyloid β; CCH, chronic cerebral hypoperfusion; CTX, cerebral cortex; DG, dentate gyrus; EDA, edaravone; HI, hippocampus; M, months; TH, thalamus.

**Figure 4 ijms-25-02945-f004:**
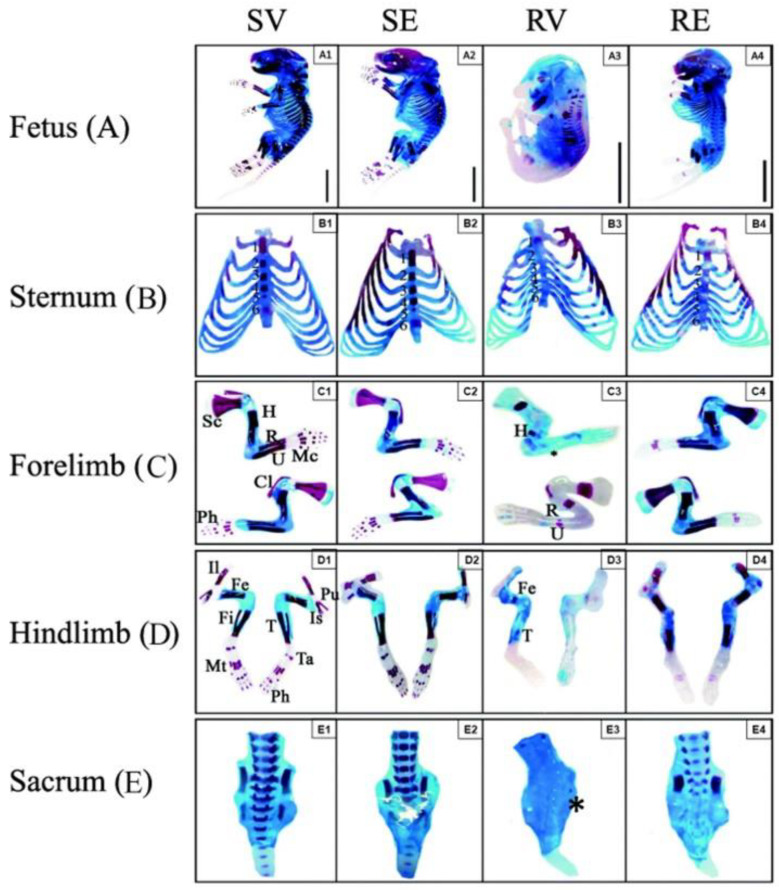
Representative photographs of the double-stained endoskeleton of fetuses and their body parts from different groups (revised by Atallah et al., 2021 [19]). (**A**–**E**) Analyses of the whole fetus and each body part showed severe loss of ossification in the RUPP-vehicle group. Notably, the lumbar and sacral vertebrae showed highly ossified central bones in the SV and SE groups (E1 and E2) and complete loss of ossification in the RV group (E3) except for a small portion of the iliac bone (asterisk). On the other hand, there was moderate improvement after edaravone injection (E4). Abbreviations: Fe, femur; Fi, fibula; H, humerus; Il, ilium; Is, ischium; Mc, metacarpus; Mt, metatarsus; Ph, phalanges; Pu, pubis; R, radius; Sc, scapula; Ta, tarsus; T, tibia; U, ulna.

**Table 1 ijms-25-02945-t001:** Therapeutic effects of edaravone in animal studies.

Animal Model	Edaravone Dose	Therapeutic Effect	References
Cerebral edema rat model created by injecting arachidonic acid into the cerebral cortex	0.1–3.0 mg/kg i.v.	Decreased brain swelling	[11]
tMCAO rat model	3.0 mg/kg i.v.	Decreased post-ischemic brain swelling	[12]
tMCAO rat model	3.0 mg/kg i.v.	Decreased the infarcts size 1 day after tMCAO	[13]
tMCAO + spontaneous hypertension rat model	3.0 mg/kg i.v.	The combination of edaravone and tPA significantly increased survival, improved motor function, and dramatically reduced hemorrhagic transformation	[14]
G93A ALS mouse model	1.0–10.0 mg/kg i.p.	Significantly slowed the motor decline of the ALS mice in rotarod test	[15]
Nrf2/G93A ALS mouse model	3.0 mg/kg i.p.	Significantly slowed the motor decline of the ALS mice in rotarod test	[16]
Chronic hypoperfusion rat model	3.0 mg/kg i.p.	Significantly improved spatial memory but not motor function	[17]
AD + chronic hypoperfusion mouse model	3.0 mg/kg i.p.	Significantly improved motor and cognitive deficits	[18]
RUPP-induced placental ischemia mouse model	3.0 mg/kg i.p.	Significantly increased fetal survival rate, fetal length, and weight. Significantly improved the histopathological structure and function of the maternal kidney	[19]

**Table 2 ijms-25-02945-t002:** Results of edaravone treatment in human clinical trials.

Study Design	Participants	Therapeutic Effect	References
Multicenter, randomized, placebo-controlled, double-blind study	Patients with acute ischemic stroke within 72 h of onset	The modified Rankin Scale was significantly improved in the edaravone group (*p* = 0.0382).	[20]
Multicenter, randomized, placebo-controlled, double-blind study	Patients with acute ischemic stroke within 48 h of onset	The edaravone-dexborneol group exhibited significantly better functional outcomes than those of the edaravone group.	[21]
Multicenter, randomized, open-labeled, single blind study	Patients with acute ischemic stroke with M1 or M2 occlusion within 3 h of onset	Good recoveries were more frequently observed in the edaravone group than in the non-edaravone group (*p* = 0.0396).	[22]
Multicenter, randomized, open-labeled, prospective study	Patients with stroke within 4.5 h of the onset	There was no difference between subjects in the early group (edaravone started before or during tPA) and the late group (edaravone started after tPA) in early recanalization.	[23]
Single-center, randomized, open-labeled, prospective study	Patients with ALS	During the 6-month edaravone treatment period, the decline in the ALSFRS-R score was significantly slower than that in the 6 months prior to administration.	[24]
Multicenter, randomized, placebo-controlled, double-blind study	Patients with ALS	The reduction of ALSFRS-R was smaller in the edaravone group than in the placebo group, but the difference was not significant.	[25]
Multicenter, randomized, placebo-controlled, double-blind study	Patients with early-stage ALS	Edaravone resulted in a significantly smaller decline in the ALSFRS-R score compared with that in the placebo group.	[26]

## Data Availability

Not applicable.

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
