# Peer review of "Update on Antioxidant Therapy with Edaravone: Expanding Applications in Neurodegenerative Diseases"

_ijms, 2024, doi:10.3390/ijms25052945_

Round 1

Reviewer 1 Report

Comments and Suggestions for Authors

Edaravone has been used in clinical practice for treatin gacute cerebral ischemia and ALS owing to its antioxidative and anti-inflammatory effects. In this article, authors introduced the unique characteristics of edaravone and other antioxidant supplement, summarize the findings to date for ischemic stroke, ALS, Alzheimer’s disease, and placental ischemia, and discuss its prospects for future therapeutic applications. It is also good that authors also mentioned their studies in this review. In my opinion, it is a nice and great job from researchers, who also work in this area effectively.

Here are my points:

-The introduction is too short. Authors should expand it.

-The Figure 1 can be drawn with appropriate drawing programs though authors took it from Nakagawa, et al. 2006.

-Figure 2 also does not look nice that authors can remove this figure as they mentioned in the text adequately.

-Is edaravone in treatment, which dosage forms? or the generic names can be added.

-The relationship of edaravone with SOD1 enzyme could be mentioned in detail.

-The edaravone takes an important place in ALS treatment. This part should be rewritten.

-In my opinion, the anti-oxidant therapy with supplements is irrelevant.

-Figure 5 can be revised that it is hard to read the text inside.

Comments on the Quality of English Language

The language is ok but some grammar mistakes can be corrected.

Author Response

Comments 1: Edaravone has been used in clinical practice for treating acute cerebral ischemia and ALS owing to its antioxidative and anti-inflammatory effects. In this article, authors introduced the unique characteristics of edaravone and other antioxidant supplement, summarize the findings to date for ischemic stroke, ALS, Alzheimer’s disease, and placental ischemia, and discuss its prospects for future therapeutic applications. It is also good that authors also mentioned their studies in this review. In my opinion, it is a nice and great job from researchers, who also work in this area effectively.

Response 1: Thank you for the supportive and thoughtful comment.

Comments 2: The introduction is too short. Authors should expand it.

Response 2: Thank you for pointing this out. We agree with this comment. Therefore, we have modified the Introduction section and added appropriate references, as below.

Introduction (Page 1, line 26 to Page 2, line 50)

The brain, which is rich in lipids and exhibits high oxygen consumption, is susceptible to damage by oxidative stress. Briefly, oxidative stress is caused by an imbalance between the production and accumulation of oxygen reactive species (ROS) in cells and tissues and the ability of a biological system to detoxify these reactive products [1]. ROS contribute to several physiological processes (e.g., cell signaling) [2] and are generated as by-products of oxygen metabolism under normal conditions. Nevertheless, environmental stressors and xenobiotics can contribute to a significant increase in ROS production, resulting in cellular and tissue damage. Oxidative stress has been implicated in neurodegenerative diseases, including amyotrophic lateral sclerosis (ALS), Parkinson's disease, Alzheimer's disease (AD), Huntington's disease, depression, and multiple sclerosis. Furthermore, it plays an important role in the pathogenesis of acute ischemic stroke [3,4]. Free radical formation and subsequent oxidative damage may be a factor in stroke severity [5]. Therefore, several antioxidants with demonstrated or predicted beneficial effects against oxidative stress and stroke have recently been reported.

Edaravone (MCI-186, 3-methyl-1 pheny-2-pyrazolin-5-one) was first established in Japan as a free radical scavenger (Figure 1). It was initially approved for the treatment of acute ischemic stroke in Japan, where it is manufactured under the brand name Radicut. It has been used in clinical practice for the treatment of acute cerebral ischemia and ALS owing to its anti-oxidative and anti-inflammatory effects. Edaravone has also been approved for use in Japan, South Korea, and the United States for the treatment of ALS. It has been approved under the brand name Radicava in the United States by the U.S. Food and Drug Administration (FDA).

References

1.           Coyle, J.T.; Puttfarcken, P. Oxidative stress, glutamate, and neurodegenerative disorders. Science 1993, 262, 689-695, doi:10.1126/science.7901908.

2.           Irani, K. Oxidant signaling in vascular cell growth, death, and survival : a review of the roles of reactive oxygen species in smooth muscle and endothelial cell mitogenic and apoptotic signaling. Circ Res 2000, 87, 179-183, doi:10.1161/01.res.87.3.179.

Comments 3: The Figure 1 can be drawn with appropriate drawing programs though authors took it from Nakagawa, et al. 2006.

Response 3:  Thank you for the suggestion. However, we have obtained the proper permission to reprint the figure, which conveys all key points. Thus, we would like to keep the current Figure 1.

Comments 4: Figure 2 also does not look nice that authors can remove this figure as they mentioned in the text adequately.

Response 4: We agree with this point. We have removed Figure 2.

Comments 5: Is edaravone in treatment, which dosage forms? or the generic names can be added.

Response 5: Thank you for raising this important point. Edaravone is available in an intravenous (IV) formulation, and it is administered as a solution through an IV infusion. In ALS treatment, edaravone is also administered by intravenous infusion. However, because long-term intravenous infusion is burdensome, an oral suspension of edaravone has been developed and is now used in clinical practice.

For clarity, we have added a description of the dosage forms of edaravone, as below.

Main Text (Page 3, lines 112–113)

Edaravone is available in an intravenous (IV) formulation, and it is administered as a solution through an IV infusion.

Main Text (Page 6, lines 235–238)

In ALS treatment, edaravone is also administered by intravenous infusion. However, because long-term intravenous infusion is burdensome for patients with ALS, an oral suspension of edaravone has been developed and is now used in clinical practice.

Comments 6: The relationship of edaravone with SOD1 enzyme could be mentioned in detail.

Response 6: Thank you for pointing it out. We have, accordingly, explained the relationship between edaravone and SOD1, as below.

Main Text (Page 5, lines 192–196)

These findings suggest that both wild-type and mutant forms of the SOD1 protein can interacts with free radicals, and this is associated with the pathological mechanisms of ALS. Edaravone probably exerts therapeutic effects not by acting directly on SOD1 but by inhibiting downstream free radicals and associated inflammatory responses.

Comments 7: The edaravone takes an important place in ALS treatment. This part should be rewritten.

Response 7: We agree. We have revised this section on ALS to emphasize this point and added an appropriate reference.

Main Text (Page 4, line 183 to Page 5 line 197)

Ito et al. performed a randomized double-blind trial of edaravone treatment in G93A mutant SOD1 transgenic mice (ALS mice). They found that edaravone treatment significantly slowed the motor decline in the mouse model of ALS [34]. In ALS model mice, we also found that 8-OHdG-positive cells appear in the perinuclear area of the spinal cord, even before the loss of motor neurons in the anterior nucleus of the spinal cord and the appearance of clinical symptoms. Furthermore, 8-OHdG expression increased in parallel with motor neuron degeneration and a loss of motor neurons [35]. In addition, recent studies using 62Cu-ATSM PET imaging have shown a significant increase in accumulation in the bilateral motor cortex and right parietal cortex in patients with ALS compared with those in controls. The degree of accumulation is correlated with disease severity in patients with ALS [36]. These findings suggest that both wild-type and mutant forms of the SOD1 protein can interact with free radicals, and this is associate with the pathological mechanisms of ALS. Edaravone probably exerts therapeutic effects not by acting directly on SOD1 but by inhibiting downstream free radicals and associated inflammatory responses.

Reference

34.         Ito, H.; Wate, R.; Zhang, J.; Ohnishi, S.; Kaneko, S.; Ito, H.; Nakano, S.; Kusaka, H. Treatment with edaravone, initiated at symptom onset, slows motor decline and decreases SOD1 deposition in ALS mice. Exp Neurol 2008, 213, 448-455, doi:10.1016/j.expneurol.2008.07.017.

Comments 8: In my opinion, the anti-oxidant therapy with supplements is irrelevant.

Response 8: We agree with this comment. We have removed the subsection (Anti-oxidant therapy with supplements).

Comments 9: Figure 5 can be revised that it is hard to read the text inside.

Response 9: Thank you for pointing this out. We have revised Figure 5 for clarity and to improve the image quality.

Figure 5 (In the revised manuscript, Figure 4)

Reviewer 2 Report

Comments and Suggestions for Authors

Oxidative stress is a phenomenon caused by an imbalance between production and accumulation of oxygen reactive species (ROS) in cells and tissues and the ability of a biological system to detoxify these reactive products. ROS can play several physiological roles (i.e., cell signaling), and they are normally generated as by-products of oxygen metabolism; despite this, environmental stressors and xenobiotics contribute to greatly increase ROS production, therefore causing the imbalance that leads to cell and tissue damage (oxidative stress). Oxidative stress is suspected to be important in neurodegenerative diseases including Lou Gehrig's disease, Parkinson's disease, Alzheimer's disease, Huntington's disease, depression, and multiple sclerosis. Furthermore, it plays an important role in acute ischemic stroke pathogenesis. Free radical formation and subsequent oxidative damage may be a factor in stroke severity. Therefore, in recent years several antioxidants have been exploited for their actual or supposed beneficial effect against oxidative stress and stroke.

The Authors provide a brief overview of the literature. This work is interesting but does not really expand upon the field. More importantly, it is not unclear if this is a systematic review. The Authors do not describe at all their approach to finding and reviewing the literature.

The presented manuscript contains very little description of research for edaravone. There is no information whether it is used only in Japan. It would be worth including data on where it is used. Moreover, the Authors should divide the presented research into behavioral tests and clinical trials, I suggest making such a review in the table. In the introduction, I lacked more detailed information about free radicals, their impact on the brain and the course or development of disorders. It is too short and does not sufficiently introduce the topic.

The next subsection - edaravone development - does not provide information how scientists obtained the described compound, where it was first used and for what purpose.

In the following subsections, there is too little information about the potential side effects of adavarone and its therapeutic potential in the context of its mechanism of action. Moreover, the Authors practically do not provide any mechanism of action other than that on free oxygen radicals. Is there no other mechanism?

Subsection 7 (Anti-oxidant therapy with supplements) is unnecessary. You should focus on adavarone because it is described too generally. Databases contain numerous information about this compound, both in the context of behavioral research and clinical trials, as well as its combined administration with other compounds. In its current form, the presented manuscript does not provide a sufficient overview of the information for the compound. Therefore, it cannot be a reliable source for other scientists. Its correction is necessary. Minor stylistic corrections are also necessary.

Author Response

Comments 1: The Authors provide a brief overview of the literature. This work is interesting but does not really expand upon the field. More importantly, it is not unclear if this is a systematic review. The Authors do not describe at all their approach to finding and reviewing the literature.

Response 1: Thank you for raising this important point. We agree with this comment and have added a description of our literature search.

Introduction (Page 2, lines 47–48)

 “Using PubMed, we searched the literature for studies related to edaravone for a comprehensive review of its development and applications to various diseases.

Comments 2: The presented manuscript contains very little description of research for edaravone. There is no information whether it is used only in Japan. It would be worth including data on where it is used. Moreover, the Authors should divide the presented research into behavioral tests and clinical trials, I suggest making such a review in the table.

Response 2: Thank you for pointing this out. We have added a more comprehensive overview of research on edaravone, including brand names and information on where it is used. In addition, as per the reviewer’s suggestion, we have added two tables (Tables 1 and 2) summarizing relevant animal experiments and clinical trials.

Main Text (Page 2, lines 69–73)

 “Phenolic compounds promote prostaglandin metabolic activity by scavenging hydroxyl radicals [8]. Edaravone, screened based on effects on prostaglandin synthesis, is a derivative of 2-pyrazolin-5-one with similar activity to that of phenol.

Reference

8.              Kuehl, F.A., Jr.; Humes, J.L.; Egan, R.W.; Ham, E.A.; Beveridge, G.C.; Van Arman, C.G. Role of prostaglandin endoperoxide PGG2 in inflammatory processes. Nature 1977, 265, 170-173, doi:10.1038/265170a0.

Main Text (Page 1, line 44 to Page 2, line 46)

It was initially approved for the treatment of acute ischemic stroke in Japan, where it is manufactured under the brand name Radicut. Edaravone has also been approved for use in Japan, South Korea, and the United States for the treatment of ALS. It has been approved under the brand name Radicava in the United States by the U.S. Food and Drug Administration (FDA).

Tables 1 and 2 (Please refer to the main manuscript)

Comments 3: In the introduction, I lacked more detailed information about free radicals, their impact on the brain and the course or development of disorders. It is too short and does not sufficiently introduce the topic.

Response 3: We agree with this comment. To address this point, we have modified the Introduction and added appropriate references, as below.

Introduction (Page 1, line 26 to Page 2, line 50)

“The brain, which is rich in lipids and exhibits high oxygen consumption, is susceptible to damage by oxidative stress. Briefly, oxidative stress is caused by an imbalance between the production and accumulation of oxygen reactive species (ROS) in cells and tissues and the ability of a biological system to detoxify these reactive products [1]. ROS contribute to several physiological processes (e.g., cell signaling) [2] and are generated as by-products of oxygen metabolism under normal conditions. Nevertheless, environmental stressors and xenobiotics can contribute to a significant increase in ROS production, resulting in cellular and tissue damage. Oxidative stress has been implicated in neurodegenerative diseases, including amyotrophic lateral sclerosis (ALS), Parkinson's disease, Alzheimer's disease (AD), Huntington's disease, depression, and multiple sclerosis. Furthermore, it plays an important role in the pathogenesis of acute ischemic stroke [3,4]. Free radical formation and subsequent oxidative damage may be a factor in stroke severity [5]. Therefore, several antioxidants with demonstrated or predicted beneficial effects against oxidative stress and stroke have recently been reported.

Edaravone (MCI-186, 3-methyl-1 pheny-2-pyrazolin-5-one) was first established in Japan as a free radical scavenger (Figure 1). It was initially approved for the treatment of acute ischemic stroke in Japan, where it is manufactured under the brand name Radicut. It has been used in clinical practice for the treatment of acute cerebral ischemia and ALS owing to its anti-oxidative and anti-inflammatory effects. Edaravone has also been approved for use in Japan, South Korea, and the United States for the treatment of ALS. It has been approved under the brand name Radicava in the United States by the U.S. Food and Drug Administration (FDA).

References

1.           Coyle, J.T.; Puttfarcken, P. Oxidative stress, glutamate, and neurodegenerative disorders. Science 1993, 262, 689-695, doi:10.1126/science.7901908.

2.           Irani, K. Oxidant signaling in vascular cell growth, death, and survival : a review of the roles of reactive oxygen species in smooth muscle and endothelial cell mitogenic and apoptotic signaling. Circ Res 2000, 87, 179-183, doi:10.1161/01.res.87.3.179.

Comments 4: The next subsection - edaravone development - does not provide information how scientists obtained the described compound, where it was first used and for what purpose.

Response 4: Thank you for pointing this out. We have added an explanation of the development of edaravone, as below.

Main Text (Page 2, lines 66–70)

 “Phenolic compounds promote prostaglandin metabolic activity by scavenging hydroxyl radicals [8]. Edaravone, screened based on prostaglandin synthesis, is a derivative of 2-pyrazolin-5-one with similar activity to that of phenol.

Reference

8.              Kuehl, F.A., Jr.; Humes, J.L.; Egan, R.W.; Ham, E.A.; Beveridge, G.C.; Van Arman, C.G. Role of prostaglandin endoperoxide PGG2 in inflammatory processes. Nature 1977, 265, 170-173, doi:10.1038/265170a0.

Comments 5: In the following subsections, there is too little information about the potential side effects of adavarone and its therapeutic potential in the context of its mechanism of action. Moreover, the Authors practically do not provide any mechanism of action other than that on free oxygen radicals. Is there no other mechanism?

Response 5: Thank you for pointing this out. We have carefully checked the literature for potential side effects of edaravone; however, additional information (beyond that reported in the review) was not available. With respect to the mechanism of action, we have added text, as below.

Main Text (Page 3, lines 104–107)

 “In addition, studies have shown that edaravone administration can increase nitric oxide-mediated vasodilation through a decrease in oxidative stress, suggesting that edaravone exerts a brain protective effect by maintaining cerebral blood flow [14,15].

References

14.         Jitsuiki, D.; Higashi, Y.; Goto, C.; Kimura, M.; Noma, K.; Hara, K.; Nakagawa, K.; Oshima, T.; Chayama, K.; Yoshizumi, M. Effect of edaravone, a novel free radical scavenger, on endothelium-dependent vasodilation in smokers. Am J Cardiol 2004, 94, 1070-1073, doi:10.1016/j.amjcard.2004.06.072.

15.         Michino, T.; Tanabe, K.; Takenaka, M.; Akamatsu, S.; Uchida, M.; Iida, M.; Iida, H. Edaravone attenuates sustained pial arteriolar vasoconstriction independently of endothelial function after unclamping of the abdominal aorta in rabbits. Korean J Anesthesiol 2021, 74, 531-540, doi:10.4097/kja.21155.

Comments 6: Subsection 7 (Anti-oxidant therapy with supplements) is unnecessary. You should focus on adavarone because it is described too generally. Databases contain numerous information about this compound, both in the context of behavioral research and clinical trials, as well as its combined administration with other compounds. In its current form, the presented manuscript does not provide a sufficient overview of the information for the compound. Therefore, it cannot be a reliable source for other scientists. Its correction is necessary. Minor stylistic corrections are also necessary.

Response 6: We agree with this comment. To improve the focus on adaravone, we have removed the subsection (Anti-oxidant therapy with supplements).

In addition, we have added six new references and Tables 1 and 2, summarizing relevant animal experiments and clinical trials.

We have carefully checked the manuscript for language and style.

Tables 1 and 2 (Please refer to the main manuscript)